# Adversarial Mixup Resynthesizers

**Christopher Beckham**[1,3], **Sina Honari**[1,2], **Alex Lamb**[1,2], **Vikas Verma**[†,1,6], **Farnoosh Ghadiri**[1,3],
**R Devon Hjelm**[1,2,5] & **Christopher Pal**[1,3,4,‡*]
[1]Mila - Québec Artificial Intelligence Institute, Montréal, Canada
[2]Université de Montréal, Canada
[3]Polytechnique Montréal, Canada
[4]Element AI, Montréal, Canada
[5]Microsoft Research, Montréal, Canada
[6]Aalto University, Finland
`firstname.lastname@mila.quebec`
[†] `vikas.verma@aalto.fi`, [‡] `christopher.pal@polymtl.ca`

## Abstract

In this paper, we explore new approaches to combining information encoded within the learned representations of autoencoders. We explore models that are capable of combining the attributes of multiple inputs such that a resynthesised output is trained to fool an adversarial discriminator for real versus synthesised data. Furthermore, we explore the use of such an architecture in the context of semi-supervised learning, where we learn a mixing function whose objective is to produce interpolations of hidden states, or masked combinations of latent representations that are consistent with a conditioned class label. We show quantitative and qualitative evidence that such a formulation is an interesting avenue of research.

## 1 Introduction

The autoencoder is a fundamental building block in unsupervised learning. Autoencoders are trained to reconstruct their inputs after being processed by two neural networks: an encoder which encodes the input to a high-level representation or *bottleneck*, and a decoder which performs the reconstruction using the representation as input. One primary goal of the autoencoder is to learn representations of the input data which are useful (Bengio, 2012), which may help in downstream tasks such as classification (Zhang et al., 2017; Hsu et al., 2019) or reinforcement learning (van den Oord et al., 2017; Ha & Schmidhuber, 2018). The representations of autoencoders can be encouraged to contain more 'useful' information by restricting the size of the bottleneck, through the use of input noise (e.g., in denoising autoencoders, Vincent et al., 2008), through regularisation of the encoder function (Rifai et al., 2011), or by introducing a prior (Kingma & Welling, 2013). Another goal is in learning interpretable representations (Chen et al., 2016; Jang et al., 2016). In unsupervised learning, learning often involves qualitative objectives on the representation itself, such as disentanglement of latent variables (Liu et al., 2017; Thomas et al., 2017) or maximisation of mutual information (Chen et al., 2016; Belghazi et al., 2018; Hjelm et al., 2019).

*Mixup* (Zhang et al., 2018) and *manifold mixup* (Verma et al., 2018) are regularisation techniques that encourage deep neural networks to behave linearly between two data samples. These methods artificially augment the training set by producing random convex combinations between pairs of examples and their corresponding labels and training the network on these combinations. This has the effect of creating smoother decision boundaries, which can have a positive effect on generalisation performance. In Verma et al. (2018), the random convex combinations are computed in the *hidden space* of the network. This procedure can be viewed as using the high-level representation of the network to produce novel training examples and provides improvements over strong baselines in the supervised learning. Furthermore, Verma et al. (2019) propose a simple and efficient method for semi-supervised classification based on random convex combinations between unlabeled samples and their predicted labels.

---

[*]Canada CIFAR AI Chair

In this paper we explore the use of a wider class of *mixing functions* for unsupervised learning, mixing in the bottleneck layer of an autoencoder. These mixing functions could consist of continuous interpolations between latent vectors such as in Verma et al. (2018), to binary masking operations to even a deep neural network which learns the mixing operation. In order to ensure that the output of the decoder given the mixed representation resembles the data distribution at the pixel level, we leverage adversarial learning (Goodfellow et al., 2014), where here we train a discriminator to distinguish between decoded mixed and unmixed representations. This technique affords a model the ability to simulate novel data points (such as those corresponding to combinations of annotations not present in the training set). Furthermore, we explore our approach in the context of semi-supervised learning, where we learn a mixing function whose objective is to produce interpolations of hidden states consistent with a conditioned class label.

## 1.1 RELATED WORK

Our method can be thought of as an extension of autoencoders that allows for sampling through mixing operations, such as continuous interpolations and masking operations. Variational autoencoders (VAEs, Kingma & Welling, 2013) can also be thought of as a similar extension of autoencoders, using the outputs of the encoder as parameters for an approximate posterior $q(\mathbf{z}|\mathbf{x})$ which is matched to a prior distribution $p(\mathbf{z})$ through the evidence lower bound objective (ELBO). At test time, new data points are sampled by passing samples from the prior, $\mathbf{z} \sim p(\mathbf{z})$, through the decoder. In contrast, our we sample a random mixup operation between the representations of two inputs from the encoder.

The Adversarially Constrained Autoencoder Interpolation (ACAI) method is another approach which involves sampling interpolations as part of an unsupervised objective (Berthelot* et al., 2019). ACAI uses a discriminator network to predict the mixing coefficient from the decoder output of the mixed representation, and the autoencoder tries to 'fool' the discriminator, making interpolated points indistinguishable from real ones. The GAIA algorithm (Sainburg et al., 2018) uses a BEGAN framework with an additional interpolation-based adversarial objective. What primarily differentiates our work from theirs is that we perform an exploration into different kinds of mixing functions, including a semi-supervised variant which uses an MLP to produce mixes consistent with a class label.

## 2 FORMULATION

Let us consider an autoencoder model $F(\cdot)$, with the encoder part denoted as $f(\cdot)$ and the decoder $g(\cdot)$. In an autoencoder we wish to minimise the reconstruction, which is simply:

$$\min_F \mathbb{E}_{\mathbf{x} \sim \mathbf{p}(\mathbf{x})} ||\mathbf{x} - g(f(\mathbf{x}))||_2 \tag{1}$$

Because autoencoders trained by input-reconstruction loss tend to produce images which are slightly blurry, one can train an adversarial autoencoder (Makhzani et al., 2016), but instead of putting the adversary on the bottleneck, we put it on the reconstruction, and the discriminator (denoted $D$) tries to distinguish between real and reconstructed $\mathbf{x}$, and the autoencoder (which is analogous to the generator) tries to construct 'realistic' reconstructions so as to fool the discriminator. Because of this, we coin the term 'ARAE' (adversarial reconstruction autoencoder). This can be written as:

$$\min_F \mathbb{E}_{\mathbf{x} \sim \mathbf{p}(\mathbf{x})} \lambda ||\mathbf{x} - g(f(\mathbf{x}))||_2 + \ell_{GAN}(g(f(\mathbf{x})), 1)$$
$$\min_D \mathbb{E}_{\mathbf{x} \sim \mathbf{p}(\mathbf{x})} \ell_{GAN}(D(\mathbf{x}), 1) + \ell_{GAN}(D(g(f(\mathbf{x}))), 0), \tag{2}$$

where $\ell_{GAN}$ is a GAN-specific loss function. In our case, $\ell_{GAN}$ is the binary cross-entropy loss, which corresponds to the Jenson-Shannon GAN (Goodfellow et al., 2014).

One way to use the autoencoder to generate novel samples would be to encode two inputs $\mathbf{h}_1 = f(\mathbf{x_1})$ and $\mathbf{h}_2 = f(\mathbf{x_2})$ into their latent representation, perform some combination between them, and then run the result through the decoder $\mathbf{g}(\cdot)$. There are many ways one could combine the two latent representations, and we denote this function $\text{Mix}(\mathbf{h_1}, \mathbf{h_2})$. Manifold mixup (Verma et al., 2018) implements mixing in the hidden space through convex combinations:

$$\text{Mix}_{\text{mixup}}(\mathbf{h}_1, \mathbf{h}_2) = \alpha \mathbf{h}_1 + (1 - \alpha)\mathbf{h}_2, \tag{3}$$

where $\lambda \in [0,1]^{(bs,)}$ is sampled from a Uniform$(0,1)$ distribution and $bs$ denotes the minibatch size.

In contrast, here we explore a strategy in which we randomly retain some components of the hidden representation from $\mathbf{h}_1$ and use the rest from $\mathbf{h}_2$, and in this case we would randomly sample a binary mask $\mathbf{m} \in \{0,1\}^{(bs \times f)}$ (where $f$ denotes the number of feature maps) and perform the following operation:

$$\text{Mix}_{\text{Bernoulli}}(\mathbf{h}_1, \mathbf{h}_2) = \mathbf{m}\mathbf{h}_1 + (1 - \mathbf{m})\mathbf{h}_2, \qquad (4)$$

where $\mathbf{m}$ is sampled from a Bernoulli$(p)$ distribution ($p$ can simply be sampled uniformly).

With this in mind, we propose the *adversarial mixup resynthesiser* (AMR), where part of the autoencoder's objective is to produce mixes which, when decoded, are indistinguishable from real images. The generator and the discriminator of AMR are trained by the following mixture of loss components:

$$\min_{F} \mathbb{E}_{\mathbf{x},\mathbf{x}' \sim \mathbf{p}(\mathbf{x})} \underbrace{\lambda||\mathbf{x} - g(f(\mathbf{x}))||_2}_{\text{reconstruction}} + \underbrace{\ell_{GAN}(g(f(\mathbf{x})), 1)}_{\text{fool D with reconstruction}} + \underbrace{\ell_{GAN}(g(\text{Mix}(f(\mathbf{x}), f(\mathbf{x}'))), 1)}_{\text{fool D with mixes}} +$$

$$\underbrace{\beta||\text{Mix}(f(\mathbf{x}), f(\mathbf{x}')) - f(g(\text{Mix}(f(\mathbf{x}), f(\mathbf{x}'))))||_2}_{\text{mixing consistency}}$$

$$\min_{D} \mathbb{E}_{\mathbf{x},\mathbf{x}' \sim \mathbf{p}(\mathbf{x})} \underbrace{\ell_{GAN}(D(\mathbf{x}), 1)}_{\text{label x as real}} + \underbrace{\ell_{GAN}(D(g(f(\mathbf{x}))), 0)}_{\text{label reconstruction as fake}} + \underbrace{\ell_{GAN}(g(\text{Mix}(f(\mathbf{x}), f(\mathbf{x}'))), 0)}_{\text{label mixes as fake}}.$$

$$(5)$$

Note that the *mixing consistency* loss is simply the reconstruction between the mix $\tilde{\mathbf{h}}_{\text{mix}} = \text{Mix}(f(\mathbf{x}), f(\mathbf{x}'))$ and the re-encoding of it $f(g(\tilde{\mathbf{h}}_{\text{mix}}))$, where $\mathbf{x}$ and $\mathbf{x}'$ are two randomly sampled images from the training set. This may be necessary as without it the decoder may simply output an image which is not semantically consistent with the two images which were mixed (refer to Section 5.2 for an in-depth explanation and analysis of this loss). Both the generator and discriminator are trained by the decoded image of the mix $g(\text{Mix}(f(\mathbf{x}), f(\mathbf{x}')))$. The discriminator $D$ is trained to label it as a fake image by minimising its probability and the generator $F$ is trained to fool the discriminator by maximising its probability. Note that the coefficient $\lambda$ controls the reconstruction and the coefficient $\beta$ controls the mixing consistency in the generator. See Figure 1 for a visualisation of the AMR model.

## 2.1 USING LABELS

While it is interesting to generate new examples via random mixing strategies in the hidden states, we also explore a supervised mixing formulation in which we *learn a mixing function that can produce mixes between two examples such that they are consistent with a particular class label*. We make this possible by backpropagating through a classifier network $p(\mathbf{y}|\mathbf{x})$ which branches off the end of the discriminator, i.e., an auxiliary classifier GAN (Odena et al., 2017).

Let us assume that for some image $\mathbf{x}$, we have a set of binary attributes $\mathbf{y}$ associated with it, where $\mathbf{y} \in \{0,1\}^k$ (and $k \geq 1$). We introduce a mixing function $\text{Mix}_{\text{sup}}(\mathbf{h}_1, \mathbf{h}_2, \mathbf{y})$, which is an MLP that maps $\mathbf{y}$ to Bernoulli parameters $\mathbf{p} \in [0,1]^{bs \times f}$. These parameters are used to sample a Bernoulli mask $\mathbf{m} \sim \text{Bernoulli}(\mathbf{p})$ to produce a new combination $\tilde{\mathbf{h}}_{\text{mix}} = \mathbf{m}\mathbf{h}_1 + (1 - \mathbf{m})\mathbf{h}_2$, which is consistent with the class label $\mathbf{y}$. Note that the conditioning class label should be semantically meaningful with respect to both of the conditioned hidden states. For example, if we're producing mixes based on the gender attribute and both $\mathbf{h}_1$ and $\mathbf{h}_2$ are male, it would not make sense to condition on the 'female' label. To enforce this constraint, we simply make the conditioning label a convex combination $\tilde{\mathbf{y}}_{\text{mix}} = \alpha\mathbf{y}_1 + (1 - \alpha)\mathbf{y}_2$ as well, using $\alpha \sim \text{Uniform}(0,1)$.

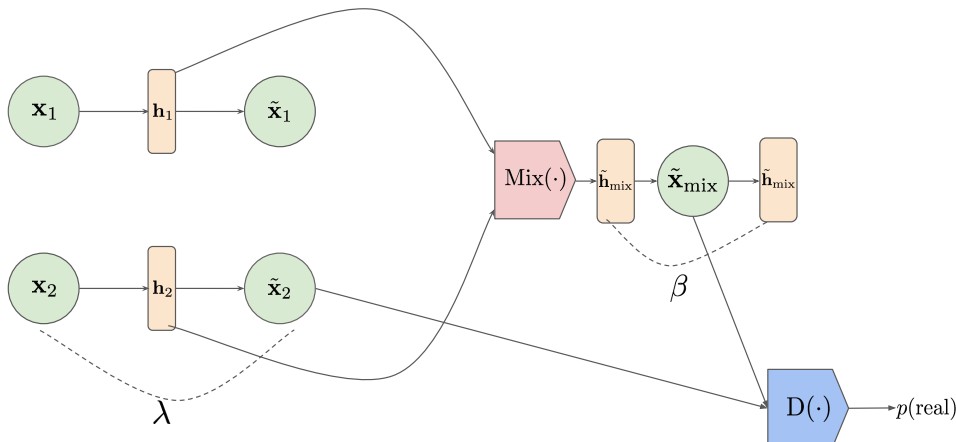

Figure 1: The unsupervised version of the adversarial mixup resynthesiser (AMR). In addition to the autoencoder loss functions, we have a mixing function Mix which creates some combination between the latent variables $\mathbf{h}_1$ and $\mathbf{h}_2$, which is subsequently decoded into an image intended to be realistic-looking and semantically consistent with the two constituent images. This is achieved through the consistency loss (weighted by $\beta$) and the discriminator.

To make this more concrete, the autoencoder and discriminator, in addition to their losses described in Equation 5, try to minimise the following losses:

$$\min_F \mathbb{E}_{\mathbf{x}_1,\mathbf{y}_2 \sim p(\mathbf{x},\mathbf{y}), \mathbf{x}_2,\mathbf{y}_2 \sim p(\mathbf{x},\mathbf{y}), \alpha \sim U(0,1)} \underbrace{\ell_{\text{GAN}}(D(g(\tilde{\mathbf{h}}_{\text{mix}})), 1)}_{\text{fool D with mix}} + \underbrace{\ell_{\text{cls}}(p(\mathbf{y}|\mathbf{g}(\tilde{\mathbf{h}}_{\text{mix}})), \tilde{\mathbf{y}}_{\text{mix}})}_{\text{make mix's class consistent}}$$

$$+ \underbrace{\beta||\tilde{\mathbf{h}}_{\text{mix}} - f(g(\tilde{\mathbf{h}}_{\text{mix}}))||_2}_{\text{consistency loss}} \quad (6)$$

$$\min_D \mathbb{E}_{\mathbf{x}_1,\mathbf{y}_2 \sim p(\mathbf{x},\mathbf{y}), \mathbf{x}_2,\mathbf{y}_2 \sim p(\mathbf{x},\mathbf{y}), \alpha \sim U(0,1)} \underbrace{\ell_{\text{GAN}}(D(g(\tilde{\mathbf{h}}_{\text{mix}})), 0)}_{\text{label mixes as fake}}$$

$$\text{where } \tilde{\mathbf{y}}_{\text{mix}} = \alpha \mathbf{y}_1 + (1-\alpha)\mathbf{y}_2 \text{ and } \tilde{\mathbf{h}}_{\text{mix}} = \text{Mix}_{\text{sup}}(f(\mathbf{x}_1), f(\mathbf{x}_2), \tilde{\mathbf{y}}_{\text{mix}})$$

Note that for the consistency loss the same coefficient $\beta$ is used. See Figure 2 for a visualisation of the supervised AMR model.

## 3 RESULTS

We use ResNets (He et al., 2016) for both the generator and discriminator. The precise architectures for generator and discriminator can be found here.[1] The datasets evaluated on are:

- UT Zappos50K (Yu & Grauman, 2014; 2017): a large dataset comprising 50k images of shoes, sandals, slippers, and boots. Each shoe is centered on a white background and in the same orientation, which makes it convenient for generative modelling purposes.
- CelebA (Liu et al., 2015): a large-scale and highly diverse face dataset consisting of 200K images. We use the aligned and cropped version of the dataset downscaled to 64px, and only consider (via the use of a keypoint-based heuristic) frontal faces. It is worth noting that despite this, there is still quite a bit of variation in terms of the size and position of the faces, which can make mixing between faces a more difficult task since the faces are not completely aligned.

---

[1]Link to the code will be added!

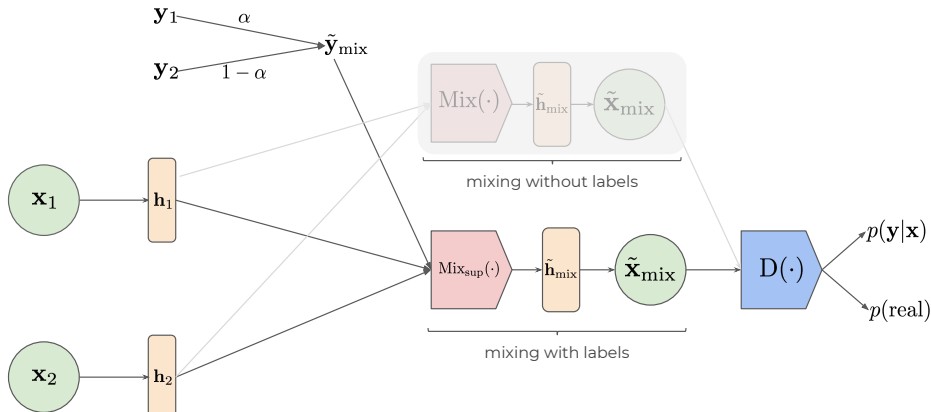

Figure 2: The supervised version of the adversarial mixup resynthesiser (AMR). The mixer function, denoted in this figure as $\text{Mix}_{\text{sup}}$, takes $\mathbf{h}_1$, $\mathbf{h}_2$ and a convex combination of $\mathbf{y}_1$ and $\mathbf{y}_2$ (denoted $\tilde{\mathbf{y}}_{\text{mix}}$) and internally produces a Bernoulli mask which is then used to produce an output combination $\tilde{\mathbf{h}}_{\text{mix}} = \mathbf{m}\mathbf{h}_1 + (1 - \mathbf{m})\mathbf{h}_2$. $\tilde{\mathbf{h}}_{\text{mix}}$ is then passed to the generator to generate $\tilde{\mathbf{x}}_{\text{mix}}$. In addition to fooling the discriminator using $\tilde{\mathbf{x}}_{\text{mix}}$, the generator also has to make sure the class prediction by the auxiliary classifier is consistent with the mixed class $\tilde{\mathbf{y}}_{\text{mix}}$. Note that in this formulation, we still perform the kind of mixing which was shown in Figure 1, and this is shown in the diagram with the component noted 'mixing without labels'.

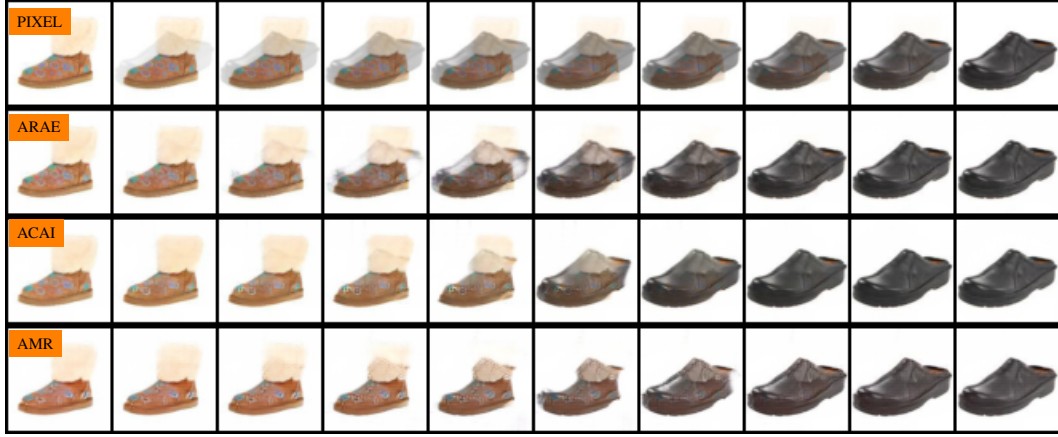

Figure 3: Interpolations between sheepskin boots and croc shoes for the Zappos dataset. For each image, from top to bottom, the rows denote: (a) linearly interpolating in pixel space; (b) the adversarial reconstruction autoencoder (ARAE); (c) adversarialy contrained autoencoder interpolation (ACAI) (Berthelot* et al., 2019) and (d) the adversarial mixup resynthesiser (AMR). For more images, consult the appendix section.

As seen in Figures 3 and 4, all of the mixup variants produce more realistic-looking interpolations than in pixel space. Due to background details in CelebA however, it is slightly harder to distinguish the quality between the different methods. Though this may not be the most ideal metric to use in our case (see discussion at end of this section) we use the Frechet Inception Distance (FID) by Heusel et al. (2017), which is based on features extracted from a pre-trained CelebA classifier, to compute the distance between samples from the dataset and ones from our autoencoders[2]. Concretely, we compute (on the validation set) two scores: the FID between validation samples and their

---

[2]This is somewhat of a misnomer because the FID implies the use of an Inception network that has been pre-trained on ImageNet. Instead, we use a ResNet which has been trained on CelebA.

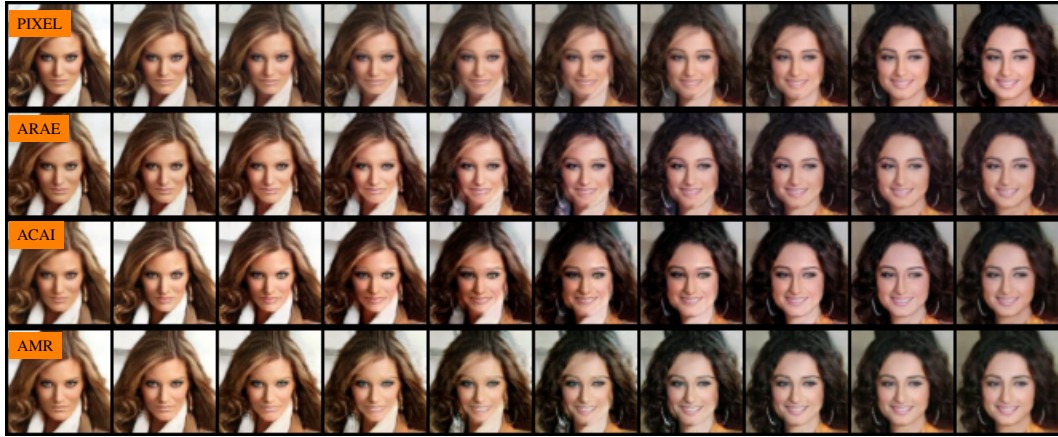

Figure 4: Interpolations between two images for the CelebA dataset, using the mixup technique (Equation 3). The left-most image is $\mathbf{x}_1$ and the right-most image is $\mathbf{x}_2$, and we create mixes between them by linearly increasing $\alpha$ from 0 to 1. For each image, from top to bottom, the rows denote: (a) linearly interpolating in pixel space; (b) the adversarial reconstruction autoencoder (ARAE); (c) adversarially constrained autoencoder interpolation (ACAI) (Berthelot* et al., 2019); and (d) the adversarial mixup resynthesiser (AMR). For more images, consult the appendix section.

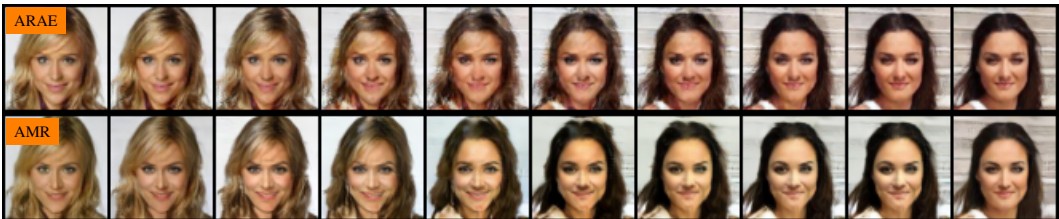

Figure 5: Interpolations between two images for the CelebA dataset, using the Bernoulli mixup technique (Equation 4). $\mathbf{m}$ is sampled from a Bernoulli(p), with zero denoting retain feature maps from $\mathbf{x}_1$ and one denoting retain feature maps from $\mathbf{x}_2$. (Top: ARAE, bottom: AMR)

reconstructions (denoted in the table as *FID(data, reconstruction)*), and the FID between validation samples and randomly sampled interpolations (denoted in the table as *FID(data, mix)*). In the latter case, we repeat this five times (over five different sets of randomly sampled interpolations) for three different random seeds, resulting in $5 \times 3 = 15$ FID scores from which we compute the mean and standard deviation. These results are shown in Tables 1 and 2 for the mixup and Bernoulli mixup formulations, respectively.

Table 1: Frechet Inception Distances computed between samples from the validation set and adversarial mixup resynthesiser (AMR) for the mixup loss (Equation 3) in the unsupervised setting. $\lambda$ and $\beta$ denote respectively the reconstruction and consistency coefficients.

| Method | $\lambda$ | $\beta$ | FID(data, mix) | FID(data, reconstruction) |
|---|---|---|---|---|
| ARAE | 20 | - | $5.94 \pm 0.30$ | $1.39 \pm 0.13$ |
| ACAI | 100 | - | $4.63 \pm 0.22$ | $0.92 \pm 0.02$ |
|  | 50 | - | $5.10 \pm 0.52$ | $1.32 \pm 0.46$ |
| AMR (ours) | 100 | 50 | $4.75 \pm 0.26$ | $1.37 \pm 0.11$ |
|  | 50 | 50 | $5.64 \pm 0.20$ | $2.59 \pm 0.09$ |
|  | 50 | 1 | $4.62 \pm 0.20$ | $1.22 \pm 0.06$ |
|  | 50 | 0 | $4.97 \pm 0.31$ | $1.26 \pm 0.12$ |

Table 2: Frechet Inception Distances computed between samples from the validation set and adversarial mixup resynthesiser (AMR) for the Bernoulli mixup (Equation 4) technique in the unsupervised setting. $\lambda$ and $\beta$ denote respectively the reconstruction and consistency coefficients.

| Method | $\lambda$ | $\beta$ | FID(data, mix) | FID(data, reconstruction) |
|--------|-----------|---------|----------------|---------------------------|
| ARAE | 20 | - | $16.64 \pm 0.59$ | $1.39 \pm 0.13$ |
| AMR (ours) | 50 | 50 | $10.36 \pm 0.84$ | $3.73 \pm 0.61$ |

Lower FID is usually considered to be better. However, FID may not be the most appropriate metric to use in our case. Because the FID is a measure of distance between two distributions, one can simply obtain a very low FID by simplying autoencoding the data, as shown in Tables 1 and 2. In the case of mixing, one situation which may favour a lower FID is if $g(\alpha f(\mathbf{x}_1) + (1 - \alpha)f(\mathbf{x}_2)) \approx g(f(\mathbf{x}_1))$ or $g(f(\mathbf{x}_2))$; in other words, the supposed mix simply decodes into one of the original examples $\mathbf{x}_1$ or $\mathbf{x}_2$, which clearly lie on the data manifold. To avoid having the mixed features $\alpha\mathbf{x}_1 + (1 - \alpha)\mathbf{x}_2$ being decoded back into samples which lie on the data manifold, we leverage the consistency loss, which is tuned by coefficient $\beta$. The lower the coefficient, the more likely that decoded mixes are projected back onto the manifold, but if this constraint is too weak then it may not necessarily be desirable if one wants to create novel data points. (For more details, see Section 5.2 in the appendix.)

Despite potential shortcomings of using FID, it seems reasonable to use such a metric to compare against baselines without any mixing losses, such as the adversarial reconstruction autoencoder (ARAE), which we indeed outperform for both mixup and Bernoulli mixup. For Bernoulli mixup, the FID scores appear to be higher than those in the mixup case (Table 1) because the sampled Bernoulli mask $\mathbf{m}$ is also across the channel axis, i.e., it is of the shape $(bs, f)$, whereas in mixup $\alpha$ has the shape $(bs, )$. Because the mixing is performed on an extra axis, this produces a greater degree of variability in the mixes, and we have observed similar FID scores to the Bernoulli mixup case by evaluating on a variant of mixup where the $\alpha$ has the shape $(bs, f)$ instead of $(bs, )$.

### 3.1 SUPERVISED CELEBA

We present some qualitative results with the supervised formulation. We train our supervised AMR variant using a subset of the attributes in CelebA ('is male', 'is wearing heavy makeup', and 'is wearing lipstick'). We consider pairs of examples $\{(\mathbf{x}_1, \mathbf{y}_1), (\mathbf{x}_2, \mathbf{y}_2)\}$ (where one example is male and the other female) and produce random convex combinations of the attributes $\tilde{\mathbf{y}}_{\text{mix}} = \alpha\mathbf{y}_1 + (1 - \alpha)\mathbf{y}_2$ and decode their resulting mixes $\text{Mix}_{\text{sup}}(f(\mathbf{x}_1), f(\mathbf{x}_2), \tilde{\mathbf{y}}_{\text{mix}})$. This can be seen in Figure 6.

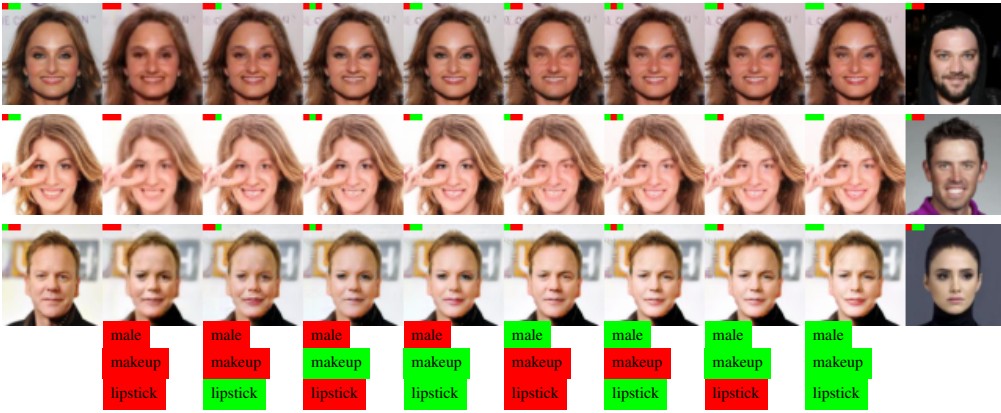

Figure 6: Interpolations produced by the class mixer function for the set of binary attributes {male, heavy makeup, lipstick}. For each image, the left-most face is $\mathbf{x}_1$ and the right-most face $\mathbf{x}_2$, with faces in between consisting of mixes $\text{Mix}_{\text{sup}}(f(\mathbf{x}_1), f(\mathbf{x}_2), \tilde{\mathbf{y}}_{\text{mix}})$ of a particular attribute mix $\tilde{\mathbf{y}}_{\text{mix}}$, shown below each column (where red denotes 'off' and green denotes 'on').

We can see that for the most part, the class mixer function has been able to produce decent mixes between the two faces consistent with the desired attributes. There are some issues – namely, the model does not seem to disentangle the lipstick and makeup attributes well – but this may be due to the strong correlation between lipstick and makeup (lipstick *is* makeup!), or be in part due to the classification performance of the auxiliary classifier part of the discriminator (while its accuracy on both training and validation was as high as 95%, there may still be room for improvement). We also achieved better results by simply having the embedding function produce a mask $\mathbf{m} \in [0, 1]$ rather than $\{0, 1\}$, most likely because such a formulation allows a greater degree of flexibility when it comes to mixing. Indeed, one priority is to conduct further hyperparameter tuning in order to improve these results. For a visualisation of the Bernoulli parameters output by the embedding function, see Section 5.3 in the appendix.

## 4 CONCLUSION

In this paper, we proposed the *adversarial mixup resynthesiser* and showed that it can be used to produce realistic-looking combinations of examples by performing mixing in the bottleneck of an autoencoder. We proposed several mixing functions, including one based on sampling from a uniform distribution and the other a Bernoulli distribution. Furthermore, we presented a semi-supervised version of the Bernoulli variant in which one can leverage class labels to learn a mixing function which can determine what parts of the latent code should be mixed to produce an image consistent with a desired class label. While our technique can be used to leverage an autoencoder as a generative model, we conjecture that our technique may have positive effects on the latent representation and therefore downstream tasks, though this is yet to be substantiated. Future work will involve more comparisons to existing literature and experiments to determine the effects of mixing on the latent space itself and downstream tasks.

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

# 5 APPENDIX

## 5.1 EXPERIMENTAL CONFIGURATION

We will provide a summary of our experimental setup here, though we also provide links to (and encourage viewers to look at) various parts of the code such as the networks used for the generator and discriminator and the optimiser hyperparameters.

We use a residual network for both the generator and discriminator. The discriminator uses spectral normalisation (Miyato et al., 2018), with five discriminator updates being performed for each generator update. We use ADAM for our optimiser with $\alpha = 2e^{-4}$, $\beta_1 = 0.0$ and $\beta_2 = 0.99$.

## 5.2 CONSISTENCY LOSS

In order to examine the effect of the consistency loss, we explore a simple two-dimensional spiral dataset, where points along the spiral are deemed to be part of the data distribution and points outside it are not. With the mixup loss enabled and $\lambda = 10$, we try values of $\beta \in \{0, 0.1, 10, 100\}$. After 100 epochs of training, we produce decoded random mixes and plot them over the data distribution, which are shown as orange points (overlaid on top of real samples, shown in blue). This is shown in Figure 7.

As we can see, the lower $\beta$ is, the more likely interpolated points will lie within the data manifold (i.e. the spiral). This is because the consistency loss competes with the discriminator loss – as $\beta$ is decreased, there is a relatively greater incentive for the autoencoder to try and fool the discriminator with interpolations, forcing it to decode interpolated points such that they lie in the spiral. Ideally however we would want a bit of both: we want high consistency so that interpolations in hidden states are semantically meaningful (and do not decode into some other random data point), while also having those decoded interpolations look realistic.

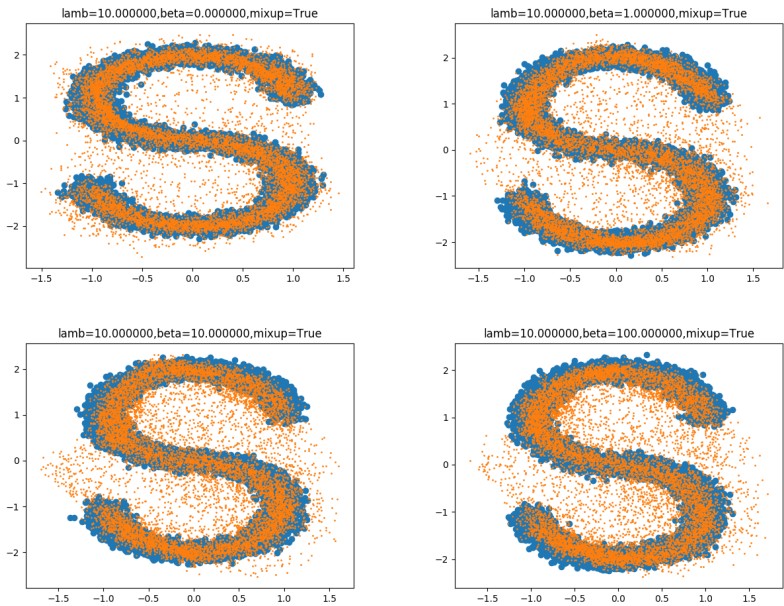

Figure 7: Experiments on AMR on the spiral dataset, showing the effect of the consistency loss $\beta$. Decoded interpolations (shown as orange) are overlaid on top of the real data (shown as blue). Interpolations are defined as $||\tilde{\mathbf{h}}_{\text{mix}} - f(g(\tilde{\mathbf{h}}_{\text{mix}}))||_2$ (where $\tilde{\mathbf{h}}_{\text{mix}} = \alpha f(\mathbf{x}_1) + (1 - \alpha)f(\mathbf{x}_2)$ and $\alpha \sim U(0, 1)$ for randomly sampled $\{\mathbf{x}_1, \mathbf{x}_2\}$)

We also compare our formulation to ACAI (Berthelot* et al., 2019), which does not explicitly have a consistency loss term. Instead, the discriminator tries to predict what the mixing coefficient $\alpha$ is, and the autoencoder tries to fool it into thinking interpolations have a coefficient of 0. In Figure 8 we compare this to our formulation in which $\beta = 0$. This is shown in Figure 8a (right figure). It appears that ACAI also prefers to place points in the spiral, although not as strongly as AMR with $\beta = 0$ (though this may be because ACAI needs to trained for longer – ACAI and AMR were trained for the same number of epochs). In Figure 8b we can see that over the course of training the consistency losses for both ACAI and AMR gradually rise, indicating both models' preference for moving interpolated points closer to the data manifold. Note that here we are only observing the consistency loss during training, and it is not used in the generator's loss.

Lastly, in Figure 9 we show some side-by-side comparisons of our model interpolating between faces when $\beta = 50$ and $\beta = 0$. We can see that when $\beta = 0$ interpolations between faces are not as smooth in terms of colour and lighting. This somewhat slight discontinuity in the interpolation

may be explained by the decoder pushing these interpolated points closer to the data manifold, since there is no consistency loss enforced.

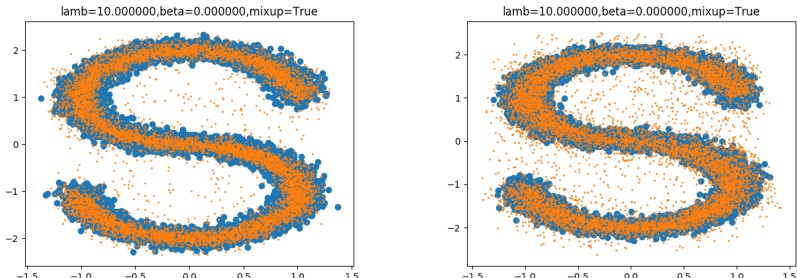

(a) Left: AMR with $\lambda = 10, \beta = 0$; right: ACAI with $\lambda = 10$ ($\beta = 0$ since ACAI does not enforce a consistency loss). AMR was trained for 200 epochs and ACAI for 300 epochs, since ACAI takes longer to converge.

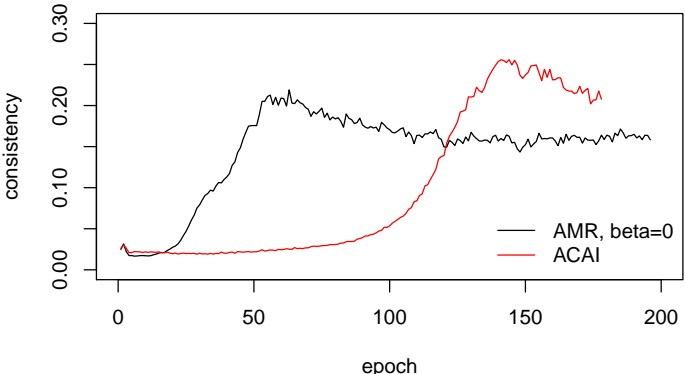

(b) Consistency loss plotted for both AMR (black) and ACAI (red).

Figure 8: Comparisons between ACAI ($\lambda = 10$) and AMR ($\lambda = 10, \beta = 0$) on the 2D spiral dataset. Top figure: decoded interpolations (shown as orange) overlaid on top of the real data (shown as blue); bottom: plot of the consistency loss $||\tilde{\mathbf{h}}_{\mathrm{mix}} - f(g(\tilde{\mathbf{h}}_{\mathrm{mix}}))||_2$ (where $\tilde{\mathbf{h}}_{\mathrm{mix}} = \alpha f(\mathbf{x}_1) + (1 - \alpha) f(\mathbf{x}_2)$ and $\alpha \sim U(0, 1)$ for randomly sampled $\{\mathbf{x}_1, \mathbf{x}_2\}$) over the course of training.

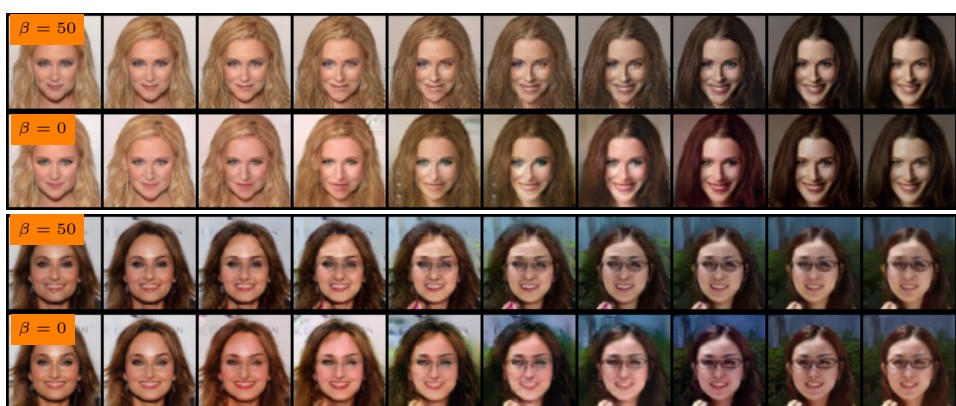

Figure 9: Interpolations using AMR $\{\lambda = 50, \beta = 50\}$ and $\{\lambda = 50, \beta = 0\}$.

## 5.3 BERNOULLI PARAMETERS FROM SUPERVISED AMR

To recap, the class mixer in the supervised formulation internally maps from a label $\tilde{\mathbf{y}}_{\text{mix}}$ to Bernoulli parameters $\mathbf{p} \in [0, 1]^K$, from which a Bernoulli mask $\mathbf{m} \sim \text{Bernoulli}(\mathbf{p})$ is sampled. The resulting Bernoulli parameters $\mathbf{p}$ are shown in Figure 10, where each row denotes some combination of attributes $\mathbf{y} \in \{000, 001, 010, \dots\}$ and the columns denote the index of $\mathbf{p}$ (spread out across four images, such that the first image denotes $\mathbf{p}_{1:128}$, second image $\mathbf{p}_{128:256}$, etc.). We can see that each attribute combination spells out a binary combination of feature maps, which allows one to easily glean which feature maps contribute to which attributes.

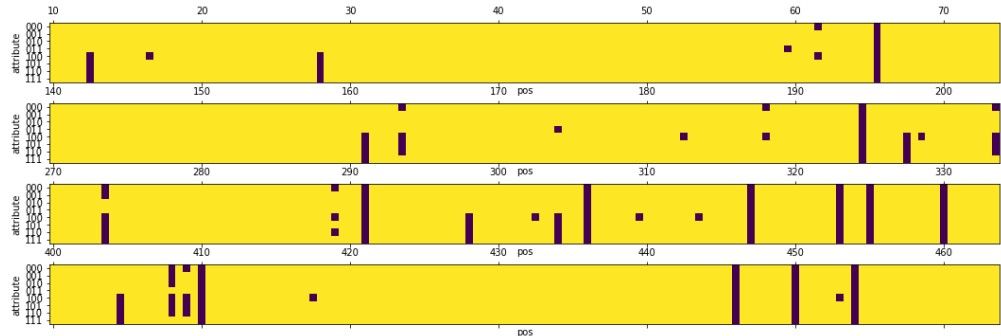

Figure 10: Visualisation of Bernoulli parameters $\mathbf{p}$ internally produced by the class mixer function. Rows denote attribute combinations $\mathbf{y}$ and columns denote the index of $\mathbf{p}$.

## 5.4 ADDITIONAL SAMPLES

In this section we show additional samples of the AMR model (using mixup and Bernoulli mixup variants) on Zappos and CelebA datasets. We compare AMR against linear interpolation in pixel space (pixel), adversarial reconstruction autoencoder (ARAE), and adversarialy contrained autoencoder interpolation (ACAI). As can be observed in the following images, the interpolations of pixel and ARAE are less realistic and suffer from more artifacts. AMR and ACAI produce more realistic-looking results, while AMR generates a smoother transition between the two samples.

- Figure 11: AMR on Zappos (mixup)
- Figure 12: AMR on Zappos (Bernoulli mixup)
- Figure 13: AMR on CelebA (mixup)
- Figure 14: AMR on CelebA (Bernoulli mixup)
- Figure 15: AMR on CelebA-HQ (Bernoulli mixup)
- Figure 16: AMR on Zappos-HQ (Bernoulli mixup)

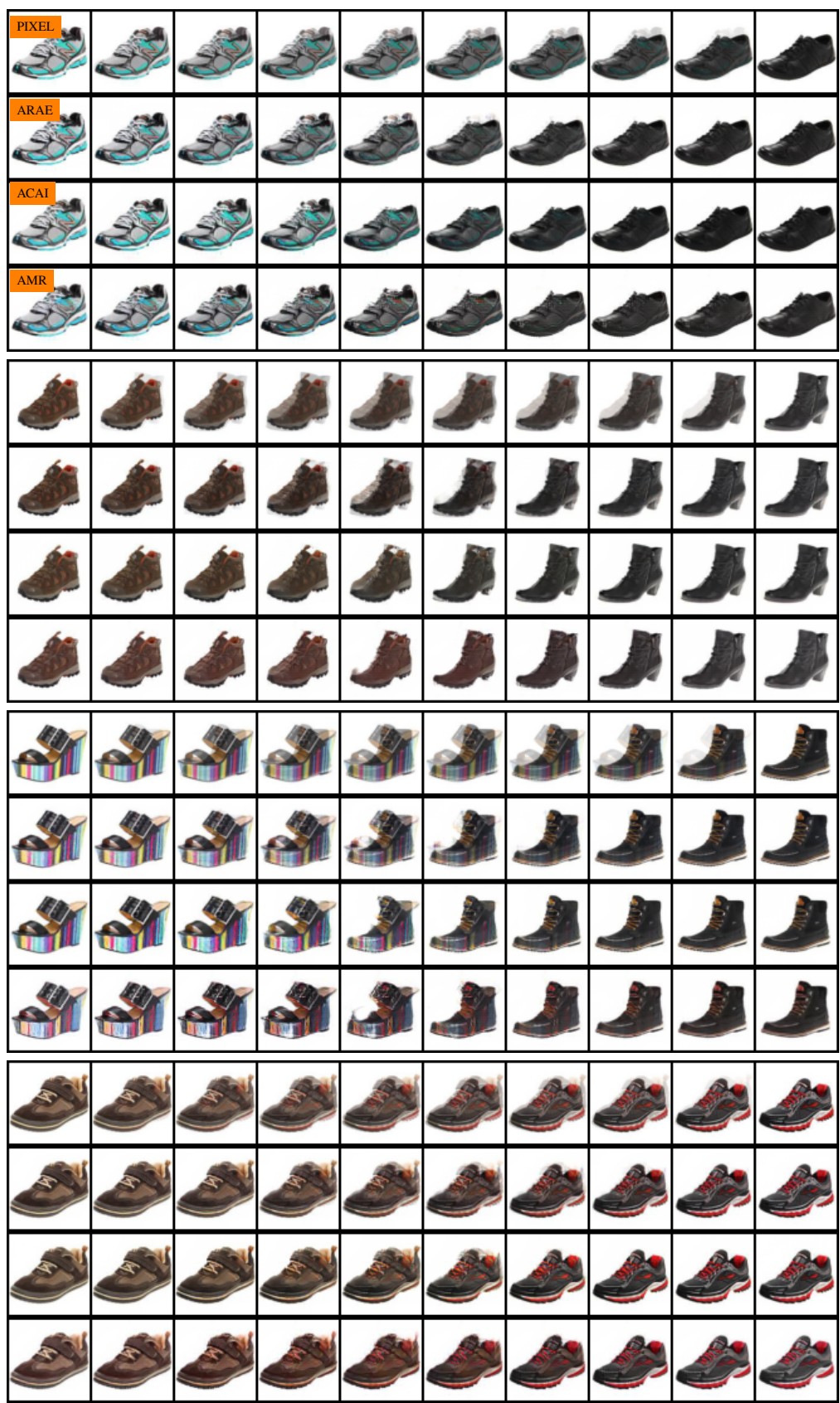

Figure 11: Interpolations between two images using the mixup technique (Equation 3). For each image, from top to bottom, the rows denote: (a) linearly interpolating in pixel space; (b) ARAE; (c) ACAI (Berthelot* et al., 2019); and (d) AMR.

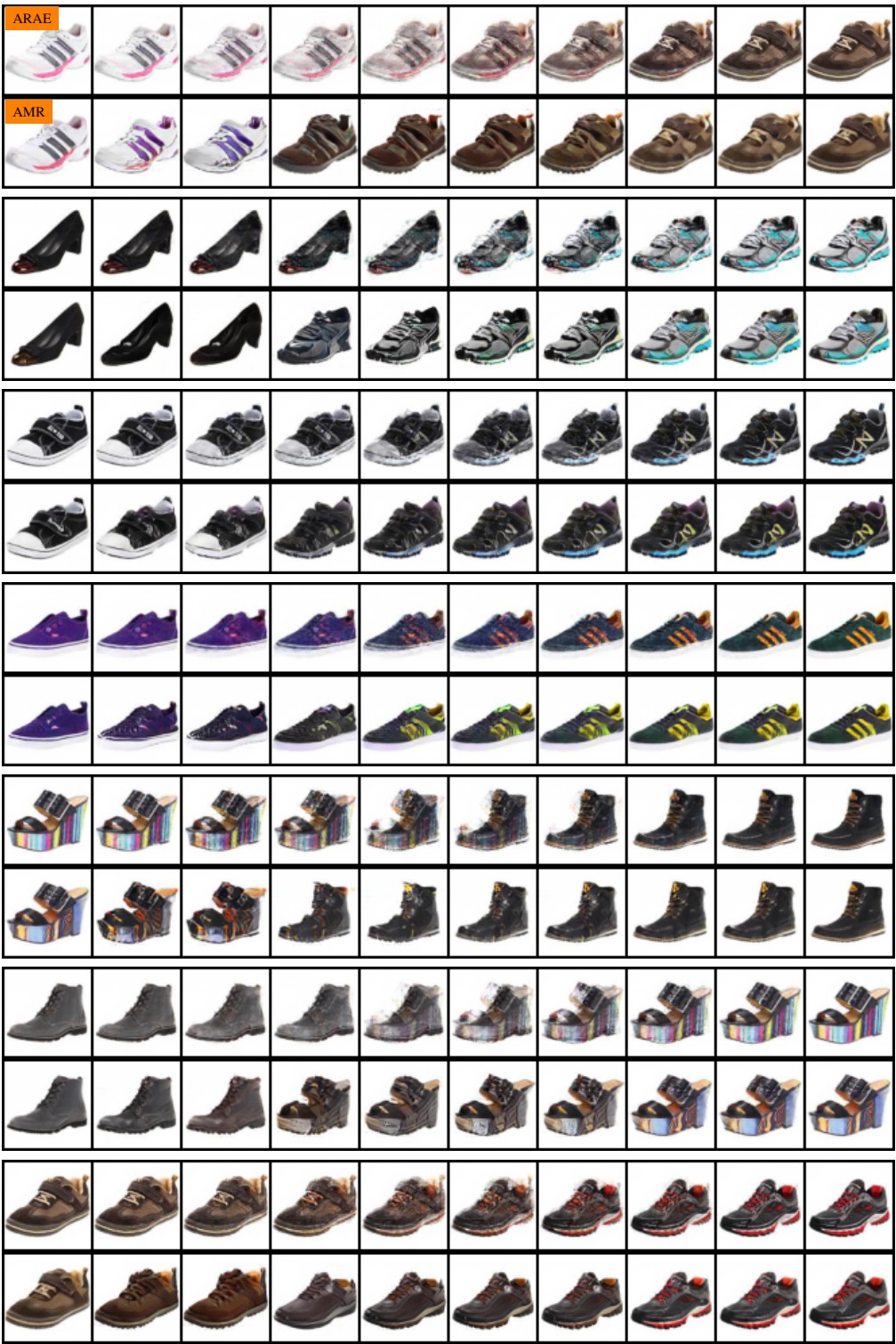

Figure 12: Interpolations between two images using the Bernoulli mixup technique (Equation 3). For each image, from top to bottom, the rows denote: (a) ARAE; and (b) AMR.

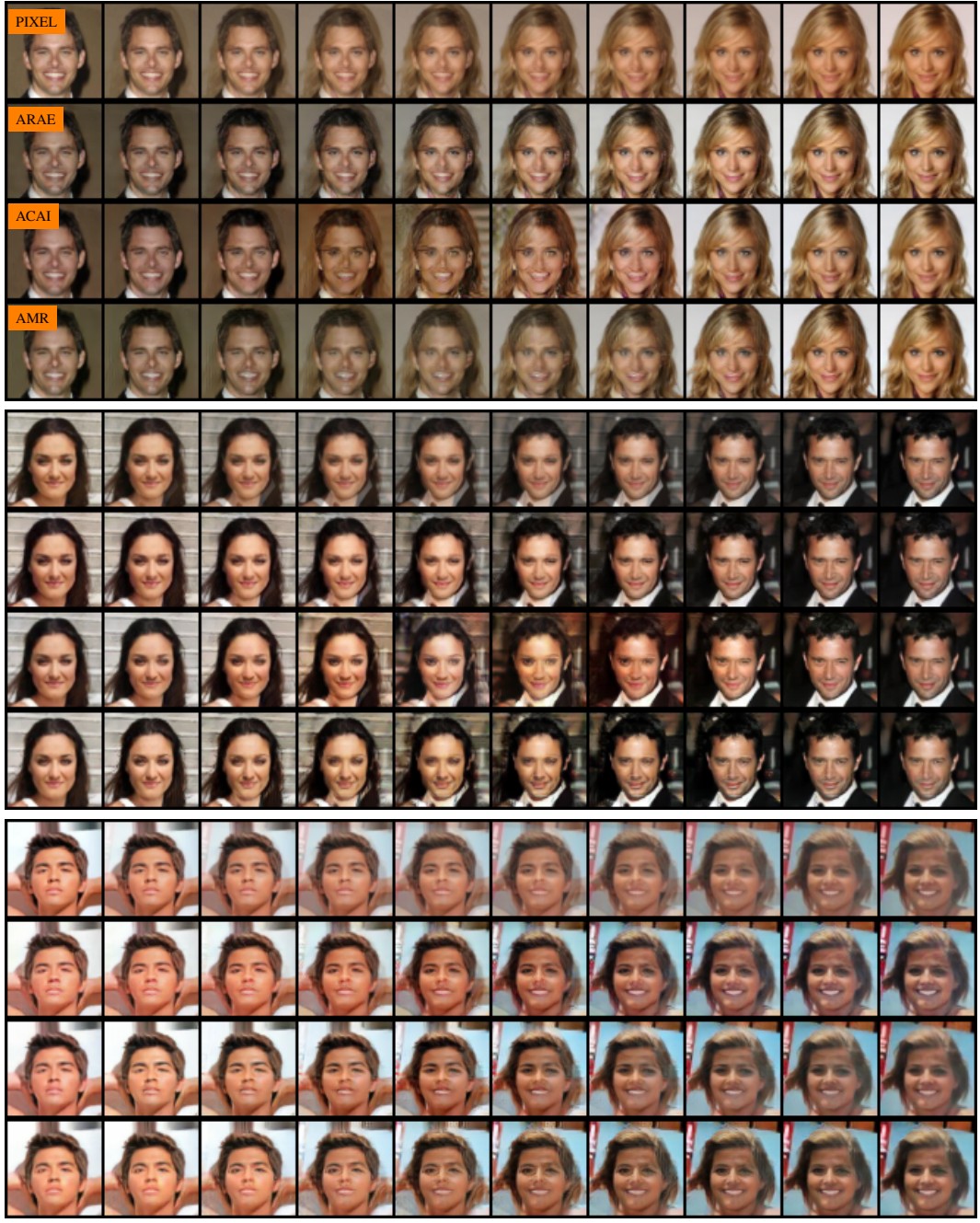

Figure 13: Interpolations between two images using the mixup technique (Equation 3). For each image, from top to bottom, the rows denote: (a) linearly interpolating in pixel space; (b) ARAE; (c) ACAI; and (d) AMR.

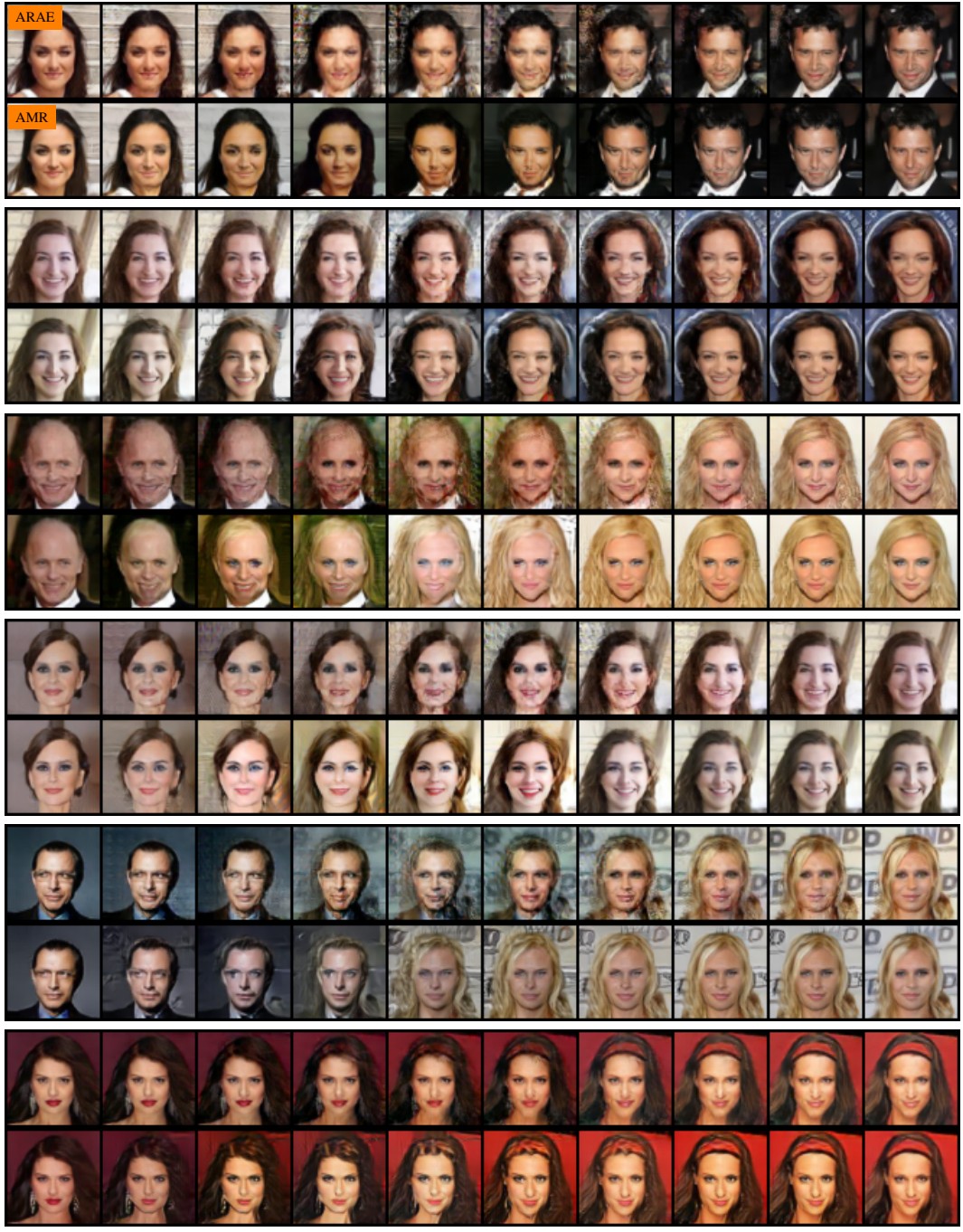

Figure 14: Interpolations between two images using the Bernoulli mixup technique (Equation 4). (For each image, top: ARAE, bottom: AMR)

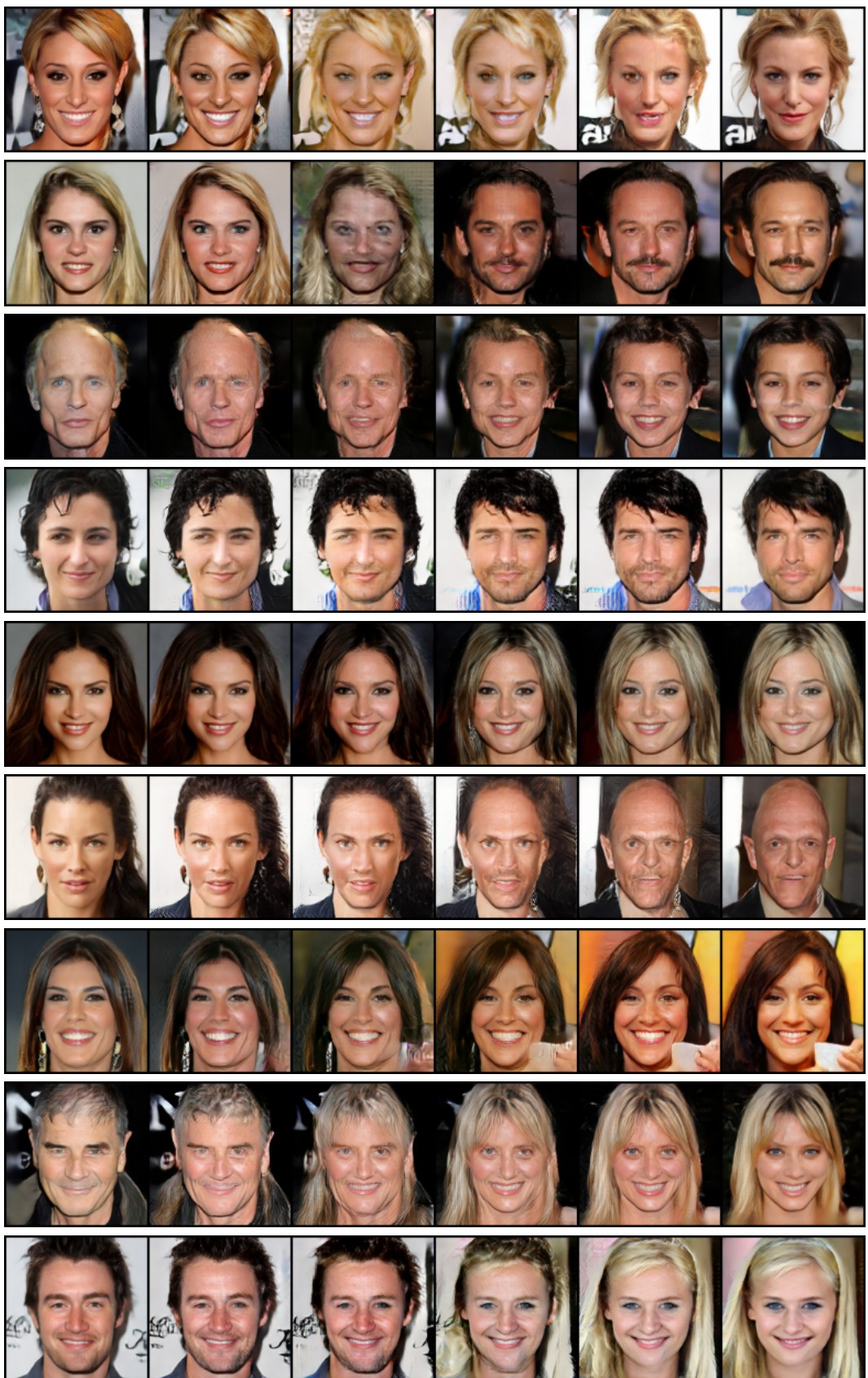

Figure 15: Interpolations between two images using the Bernoulli mixup technique (Equation 4). Each row is AMR.

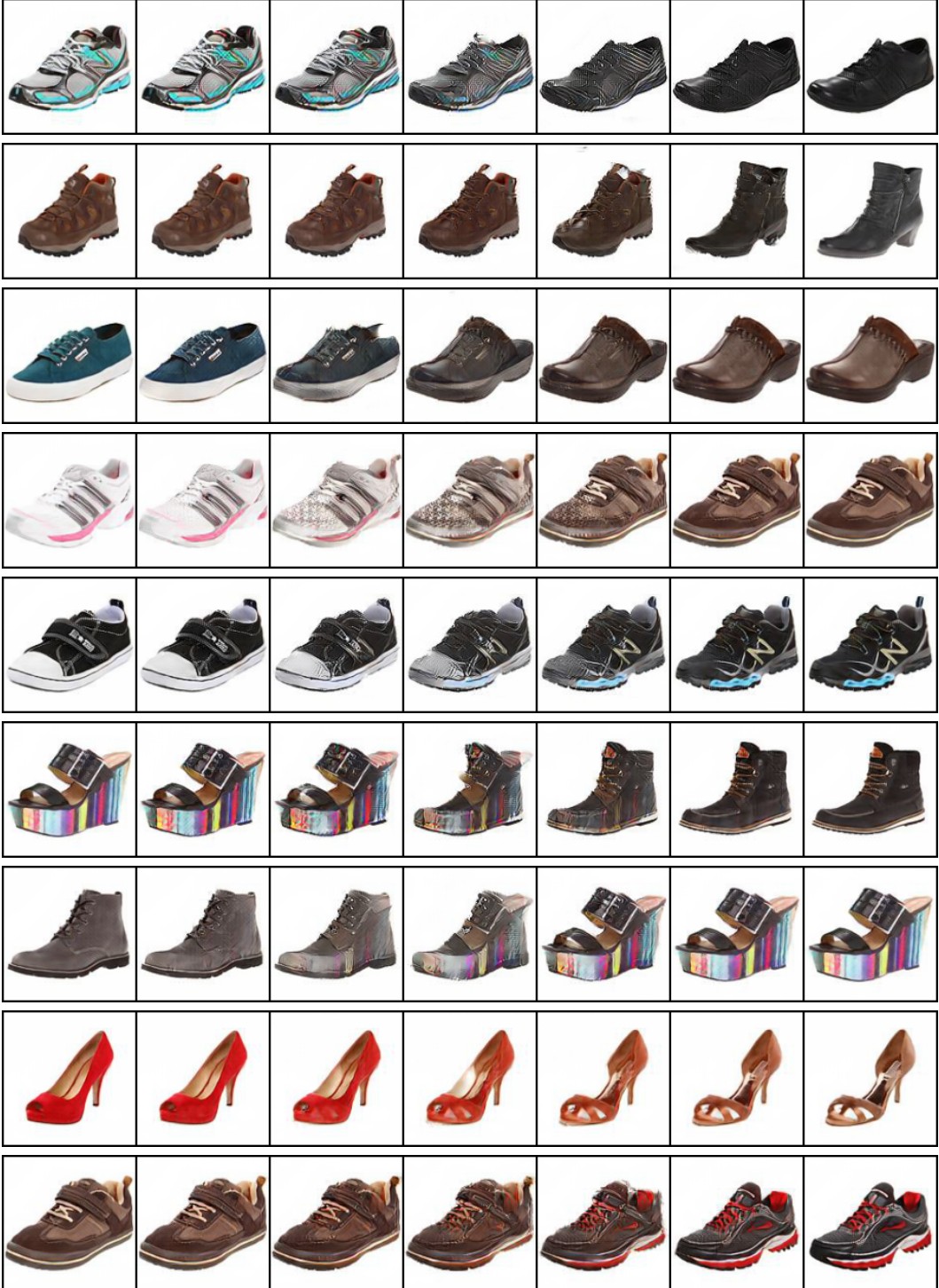

Figure 16: Interpolations between two images using the Bernoulli mixup technique (Equation 4). Each row is AMR.

