# OpenReview forum: "Adversarial Mixup Resynthesizers"
_ICLR.cc/2019/Workshop/DeepGenStruct — DeepGenStruct 2019_

### Official Review · AnonReviewer1 · 2019-04-16
**Strong results in need of a little theory**

**Rating:** 4
**Confidence:** 1

**Review:**

The authors propose to construct a generative model by training an autoencoder with a structured latent space. In this latent space, new datapoints can be generated by interpolating between existing datapoints, either linearly or by masking and replacing. They demonstrate that this model performs well at interpolating between data and that it can make use of supervised data attributes.

My main lingering question after reading the paper is about how exactly the method works theoretically. That is, is there a single objective which this method optimizes? Exactly what properties should we expect the latent space to have?

Minor note: On the first line of text after Equation 3, it says lambda where I believe it was meant to say alpha.

Pros:
- Good results
- Strong empirical exploration

Cons:
- Limited theory and complicated objective function sheds little light on the mechanism of the method's success

---

### Official Review · AnonReviewer2 · 2019-04-18
**OK, but not good enough yet**

**Rating:** 2
**Confidence:** 2

**Review:**

This paper proposes to improve representations learned by autoencoders by applying ideas from adversarial learning and mixup.

I think this is an interesting area of research, however I think the current draft is not ready for acceptance yet.

Too many moving parts in equations (5) and (6)  which makes it difficult to understand why the method works.

Some of the design choices seem arbitrary without any clear discussion:
e.g. why Bernoulli mixup instead of usual mixup?
The mixing consistency loss in (5) is not explained clearly, and seems like a hacky fix.

I was expecting to see at least some discussion of why this loss is interesting from a theoretical perspective and how it’s different from existing methods for regularizing auto-encoders. Difference from closely related work (e.g. Berthelot et al. 2019) is not clearly discussed. The authors defer to supplementary material for key details and even the discussions in sections 5.2 and 5.3 of supplementary material do not convincingly address these questions.

Results in Table 1:  proposed method doesn’t seem to be significantly outperform ACAI.
Results in Table 2: why no comparison to ACAI?

---

### Decision · Program_Chairs · 2019-04-19
**Acceptance Decision**

**Decision:**

Accept

**Comment:**

The paper's results are interesting but the reviewers note that the loss is not well motivated theoretically. It would be good to explore this aspect further.